# Capitulum Development and Gametophyte Ontogeny: Histological Insight into the Reproductive Process of a Hexaploidy Population of *Solidago canadensis* in China

**DOI:** 10.3390/plants11152073

**Published:** 2022-08-08

**Authors:** Yu Zhang, Fei Cao, Sheng Qiang

**Affiliations:** Weed Research Laboratory, Nanjing Agricultural University, Nanjing 210095, China

**Keywords:** *Solidago canadensis* L., invasive species, sexual reproduction, capitulum development, mega and microsporogenesis, female and male gametophytes

## Abstract

*Solidago canadensis* L., native to North America, has become a troublesome invasive plant worldwide due to its strong sexual reproductive capacity. Although there have been studies on some stages of sexual reproduction, there has been no systematic description of the process. In this study, we observed capitulum development, the occurrence of megasporogenesis and microsporogenesis, and embryo development using a scanning electron microscope. The results showed that there was a close relationship between the length of the capitulum bud and the stage in the reproductive process. Capitulum development appeared when the length of the capitate inflorescence was less than 1.73 ± 0.08 mm. The meiosis of microspores occurred when the length of the capitate inflorescence ranged from 2.20 ± 0.07 mm to 3.50 ± 0.10 mm, and mature pollen grains and embryo sacs formed when the length of the capitate inflorescence was greater than 5.15 ± 0.14 mm. Based on the available information, a reproductive calendar showing the key reproductive events from capitulum development to seed formation has been prepared. These processes may be related to its inherent temperature adaptation and non-synchronization of flowering, which may avoid embryo abortion during embryo development and consequently as a key step for its successful invasion in China. These results open up new horizons for effective prevention and control of spread in the future.

## 1. Introduction

*Solidago canadensis* L. is a perennial herb of the Asteraceae family. Native to North America [1], it has invaded all over Europe, most of Asia, Australia, and New Zealand [2,3,4]. In 1935, *S. canadensis* was first introduced into Shanghai and Nanjing as a garden flower, then escaped into the wild and became a malignant weed [5]. It is an autopolyploid species [6] and comes in three main cytotypes in its native ranges, including diploids (2*n* = 18), tetraploids (2*n* = 36), and hexaploids (2*n* = 54) [7]. Only tetraploids and hexaploids have been found in the field in China [2,6,8]. Due to its polyploidization-driven strong vegetative and sexual reproductive capacity and superior competitiveness, it tends to establish and form a single dominant community in its invasive range [2,6,9,10]. Invasion by this species destroys the structure and function of the ecosystem, threatens local biodiversity, and could lead to the local extinction of native species [5,6]. Research into the invasion mechanism of *S. canadensis* is currently receiving increasing attention in a bid to control its spread and effectively mitigate its harmfulness.

*S. canadensis* presents the typical characteristics of an invasive plant, namely rapid growth and large biomass [6,11,12] These characteristics are expressed in strong competitive interactions when *S. canadensis* is grown with local species and are of great significance for population diffusion and colonization [6]. However, in the colonization of new sites, sexual reproduction and the production of numerous widely dispersed seeds are of crucial importance [13,14]. Several studies have found that the capacity of *S. canadensis* to form a larger- and longer-lived soil seed-bank than its competitors probably contributes to its significant colonization abilities [15]. Moreover, its achenes with pappus can travel long distances, transported mainly by wind, creating multiple “satellite” populations [16] that eventually coalesce into a single dominant population [6,9]. Considering the importance of sexual reproduction in the invasion and spread of *S. canadensis*, many studies have been carried out on the mechanisms of its sexual reproduction. Pullaiah [17] described the development of female and male gametophytes. Smith and Johnson [18] used the complete transparency of ovule development to observe early ovule development, megasporogenesis, and female gametophyte development. Cheng [2] concluded that ploidy-dependent fertility appears to play a key role in the hot-summer climate in an introduced range in the northern hemisphere through both pre-adaptation and the rapid post-introduction adaptive evolution of delayed flowering and improved heat tolerance during embryo development [2]. However, there has been no comprehensive study on capitulum development and embryo development in *S. canadensis*. A systematic description of the sexual reproduction process in *S. canadensis* could provide important information for predicting the extent of its invasion.

In addition, the spread of plants is affected by phenology, especially the plant’s flowering date. Plants pollinated by insect flower when pollinators are available and environmental conditions are suitable for plant growth and reproduction [19]. The flowering period of *S. canadensis* in its native range in Canada was August (http://www.ontariowildflowers.com/ (accessed on 23 May 2022), but the flowering time shifted to October after its introduction to China [2]. A study on the seasonal synchronization of flowering of *S. canadensis* could provide a good opportunity for the effective control of this species.

In this study, the sexual reproduction process of capitulum development, the occurrence of megasporogenesis and microsporogenesis, the development of male and female gametophytes, and the embryonic development of a hexaploid *S. canadensis* population CN04 were studied systematically, with particular emphasis on the relationship between the internal development process and external morphological characteristics. The purpose is to reveal the complete process of sexual reproduction of the species after invasion and colonization under the climatic conditions of the invasion site and to provide a theoretical basis for preventing and controlling the spread of the species.

## 2. Results

### 2.1. Panicle Morphology at Different Development Stages and the Corresponding Stage of Capitulum Development

The panicle and the capitulum of *S. canadensis* at different development stages are shown in Figure 1. In late August, the leaves at the top of the plant began to change significantly, from the vegetative growth stage, which has broad leaves (Figure 1a), to the narrow leaves closing together into the shape of a rosette (Figure 1b). The axillary buds in the upper leaves began to sprout and developed into the first branch of the panicle. The main and other branches continued to elongate over time. In early September, the tops of the main and subsidiary branches of *S. canadensis* first differentiated into short branches. In mid-September, the green capitulum buds were spherically dense in the axillary buds on the short branches (Figure 1c). When the panicle had formed, the secondary branches in the middle and lower parts of the first branch were observed (Figure 1d). With the development of panicles, the capitulum of *S. canadensis* gradually adopted a scorpion shape on the inflorescence branches and then combined into a large panicle (Figure 1e,f). The morphological characteristics following the development of the capitulum are shown in Figure 1i–p. The process of capitulum development on each lower branch was the same as that on the top of the main branch, but the start time was later than that on the top. In this study, capitulum development on the bottom branch was about one week later than on the top.

According to the morphological characteristics, the capitulum development of *S. canadensis* was roughly divided into seven stages:

The pre-differentiation stage: when the length of the capitulum is less than 0.31 ± 0.03 mm, a dense globular protrusion appears in the axillary buds at the top of the plant (Figure 1b). The protrusion is the growth point of the capitulum (Figure 2a,k), and the growth point is conical.

The periclinium differentiation stage: When the length of the capitulum is between 0.31 ± 0.03 mm and 0.46 ± 0.04 mm, the growth point of the capitulum enlarges like a hemispherical receptacle (Figure 2b,l), and the involucre differentiates around it in a way that corresponds to the plant morphology (Figure 1c).

The floret primordium differentiation stage: When the length of the capitulum is between 0.46 ± 0.04 and 0.67 ± 0.03 mm, the floret primordium differentiates into the shape of round granules on the hemispherical receptacle. The middle protrusion is the tubular flower primordium, surrounded by the ligulate flower primordium (Figure 2c,m). The protrusion of the tubular flower primordium is significantly larger than the ligulate flower primordium.

The corolla primordium differentiation stage: When the length of the capitulum is between 0.67 ± 0.03 and 0.91 ± 0.05 mm, the periphery of the floret primordium bulges and the middle part recedes (Figure 2d,n). The bulge is the corolla primordium.

The calyx and stamen primordium differentiation stage: When the length of the capitulum is between 0.91 ± 0.05 and 1.22 ± 0.07 mm, the corolla primordium of the tubular flower splits into a star shape with 5 lobes. In the inner corolla, the stamen primordium begins to differentiate under each horn (Figure 2e,o). In the relative position of each horn of the outer layer of the corolla, the first group of calyx primordia appears (Figure 2e). As the petal primordium gathers to the middle, the stamen primordium gradually becomes larger and longer (Figure 2f). The second group of calyx primordia is opposite to the petal primordia, alternating with the first group of calyx primordia (Figure 2g). The protuberance of the ligulate flower is cylindrical. The inner layer of the corolla does not change significantly, and the calyx primordium appears in the outer layer (Figure 2g). 

The pistil primordium differentiation stage: When the length of the capitulum is between 1.22 ± 0.07 and 1.40 ± 0.04 mm, the petal primordium of the tubular flower elongates and gathers to the middle (Figure 2h,p). The pistil primordium begins to differentiate in the inner layer of the stamen primordium, and the ligulate flower gradually elongates its calyx primordium. The pistil primordium in the corolla begins to differentiate (Figure 2i,q).

The complete differentiation stage: When the length of the capitulum is between 1.40 ± 0.04 and 1.73 ± 0.08 mm, with further development of capitulum, the differentiation of each part of the flower is completed. The petals of the tubular flower are completely gathered in the middle, covering the stamen and the pistil. The calyx, corolla, and pistil of the ligulate flower continue to elongate. Finally, the calyx degenerates into the pappus (Figure 2j), and the nucellus primordium of the ovary can be seen from the longitudinal section (Figure 2r). 

In this study, the capitulum development of *S. canadensis* started in mid-September, when the average temperature dropped rapidly to 22.5 °C and stayed there for about 7 days. When the capitulum development was complete, the secondary branches of the inflorescence emerged from the axillary bud and grew vertically up the main branch in the form of short sticks (Figure 1d). The capitulum also elongated, changing from an initially spherical shape (Figure 1i) to an olive shape with a large middle and small ends (Figure 1j).

### 2.2. Microsporogenesis and Male Gametophyte Development

When the length of the capitulum was between 1.73 ± 0.08 and 1.90 ± 0.06 mm, the archesporium differentiated under the pellicle of the anther (Figure 3a) and went through periclinal division (Figure 3b). The outer surface became the primary wall cells, and the inside became the primary sporogenous cells (Figure 3c). The primary parietal cells and primary sporogenous cells continued to develop. When the length of the capitulum was 2.20 ± 0.07 mm (Figure 1k), microsporocytes formed. Four-layer walls could be observed, and these included the epidermis, endothecium, middle layer, and tapetum layer (Figure 3d). When the length of the capitulum was between 2.20 ± 0.07 and 3.22 ± 0.09 mm (Figure 1l), the meiosis stage of microsporocytes occurred (Figure 3e–j). The microsporocytes were divided into two one-nucleate cells by the first meiosis (Figure 3g), and they formed a tetrahedral tetrad after the second meiosis (Figure 3j). The microspore tetrahedral tetrad was surrounded by callose, arranged in a tetrahedric shape (Figure 3j). At last, the four microspores were separated, and the meiosis of the microsporocytes lasted about four days. The cytoplasmic division of microsporocyte meiosis occurred simultaneously. When the length of the capitulum was between 3.22 ± 0.09 and 5.15 ± 0.14 mm (Figure 1m), the microspores were developing into mature pollen. After the microspores were released from the tetrads, they formed a distinct wall (Figure 3k). The cytoplasm of the microspores vacuolated to form a large central vacuole (Figure 3l). During this process, the volume increased rapidly, and the cell nucleus was compressed by the vacuole to one side next to the cell wall. Then the cells underwent mitosis to form two-celled pollen (Figure 3m), and the germ cells further underwent mitosis to form three-celled pollen (Figure 3n). When the pollen matured, the pollen sac ruptured, and the pollen scattered (Figure 1n and Figure 3o).

*S. canadensis* has an amoeboid tapetum. When the microsporocytes went through the second meiosis, the tapetum layer began to dissolve (Figure 3h). When the microspores were separated from the tetrad, the protoplasts of the tapetum layer began to flow into the anther cell. Throughout the development of the microspore, the periplasmodium could be seen in the gaps between the microspores (Figure 3k–n).

At the same time that the archesporium developed to the tetrahedral stage (Figure 3j), the branches of the panicle moved from the previous vertical growing direction while continuing to elongate (Figure 1e). The capitulum continued to elongate; its upper end became sharp, and the lower end expanded into an oval shape (Figure 1k). With further elongation, the capitulum became conical (Figure 1l). At this time, the microspores were undergoing meiosis. With the development of the male gametophyte, the upper end of the capitulum gradually became round, and the color of the inflorescence changed from the early tender green to yellowish green (Figure 1m). At this time, the plant exhibited a completely developed panicle (Figure 1f). When the mature pollen grains were formed (Figure 3o), the yellow flower crown was exposed, with a cylindrical capitulum (Figure 1g,n). In this study, the process from the occurrence of the archesporium to the release of mature pollen grains lasted about 18 days. During flowering, the whole panicle was golden yellow (Figure 1g). The upper part of the capitulum blossomed, as well as the corolla of the tubular flower, and the stamens, whose anthers were chapped, stretched from the inflorescence (Figure 1o). After about 8 days, the capitulum withered (Figure 1h,p). The number of tubular flowers in the capitulum of the *S. canadensis* was three to five, and the pollen grains in the same capitulum matured within a similar time-frame. However, the corollas of the tube flowers did not all blossom at the same time (Figure 1g), so the pollen was not released simultaneously.

### 2.3. Megasporogenesis and Female Gametophyte Development

Soon after the microsporogenesis took place, when the length of the capitulum was between 2.20 ± 0.07 mm and 5.15 ± 0.14 mm, megasporogenesis would happen. The lower part of the ovary of *S. canadensis* had one anatropous ovule, with a single integument and a thin nucellus (Figure 4a). An archesporium differentiated under the epidermis of the nucellus (Figure 4b), and this functioned directly as a megasporocyte (Figure 4c). The meiosis of the megasporocyte lasted about five days and produced a linear megaspore tetrad (Figure 4d). The embryo sac was a typical polygonum type, and the three megaspores at the micropylar end had degenerated. The megaspore at the chalazal end became functional and developed into a one-nucleate embryo sac (Figure 4e). The one-nucleate embryo sac underwent a nuclear division to form a two-nuclear embryo sac (Figure 4f). When the volume of the two-nucleate embryo sac increased, each nucleus underwent mitosis to form a four-nucleate embryo sac (Figure 4g). Mitosis then took place again and formed an eight-nucleate embryo sac (Figure 4h). After further development, a mature embryo sac (Figure 4i) was formed, and multiple antipodal cells were formed inside, in a linear arrangement.

In the case of the megasporocyte, the nucellus was surrounded by the integument, and the epidermal cells adjacent to the nucellus were densely arranged (Figure 4c). During the meiosis of the megasporocyte, these epithelial cells developed into the mature tapetum layer of the integument (Figure 4d). At the one-nuclear embryo sac stage, the nucellus epidermis degenerated, and the endothelium of the integument directly covered the embryo sac (Figure 4e).

### 2.4. Flowering and Pollination Stage

*S. canadensis* is an insect-pollinated plant. Following the same order as capitulum development, the capitulum on the top of the plant blossomed first, followed by the lower part. When the middle and lower parts were in full bloom, the top capitulum had faded (Figure 1h). In this study, the full-bloom stage of the CN04 population was in mid-October.

### 2.5. Embryo Development

The embryonic development of *S. canadensis* was identical to that of most dicotyledonous plants, from a globular embryo (Figure 4j) to a heart-shaped embryo (Figure 4k), to a torpedo-shaped embryo (Figure 4l), to finally to a mature embryo (Figure 4m). Endosperm formation is the nuclear type. At the stage of the globular embryo, endosperm cells filled the entire embryo sac (Figure 4j). At the heart-shaped embryo stage, the endosperm cells around the embryo began to disintegrate and were gradually absorbed by the developing embryo (Figure 4k). During the mature stage, the endosperm was almost completely absorbed, leaving only one layer of endosperm coat (Figure 4m). The suspensor was visible in the early stage of the heart-shaped embryo but completely degenerated during the torpedo-shaped embryo and mature embryo stages (Figure 4l,m). Embryonic development began in middle and late October and produced mature seeds in early November. According to meteorological data, sunny days with a wind greater than 8 m/s often occurred in November in Nanjing over the past 10 years, and this has been beneficial to the widespread dispersal of the mature seeds of *S. canadensis*.

### 2.6. Capitulum Morphology at Different Development Stages of Differentiation and the Corresponding Stage of Microspore and Megasporogenesis

According to our observation, the capitulum development of *S. canadensis* began with a time gap of about one week, while the duration of capitulum development in the subsequent stages was the same. The developmental periods of capitulum development, microspore, and megasporocyte meiosis were seven days, four days, and five days, respectively. Figure 5 records the duration of capitulum development, the occurrence of megasporogenesis and microsporogenesis, and their correspondence with capitulum morphology or the length of capitulum. In general, within the flower of *S. canadensis*, the stamen appeared and developed earlier than the pistil, but they matured around the same time before flowering.

## 3. Discussion

Since 1930s, *S. canadensis* has mainly invaded the eastern region of China but is now spreading to many other provinces such as Henan, Liaoning, Sichuan, Hunan, Guangdong, Guangxi, and Yunnan [5]. The extraordinary rate of the invasion by this species is strongly related to its sexual reproductive ability [13]. The seeds of *S. canadensis* are spread over long distances by human disturbance, wind, birds, and animals. First, “satellite” populations form, and then the gaps are filled by the vegetative propagation of underground stems [9]. Eventually, a single dominant community will form. As for its distribution, climate is a major factor [20,21], and the adaptation of an alien species to the climate facilitates its successful invasion. Furthermore, the reproductive process is more sensitive to the environment than the vegetative growth. At the initial stage of the establishment of *S. canadensis*, it tends to adapt its sexual reproduction mechanism to complete its settlement. Sexual reproduction is thus extremely important to its successful invasion [13].

The flowering time of *S. canadensis* in its native range, Ontario, Canada, is August; capitulum development takes place in July, and the seeds mature in early September [2]. The mean maximum and minimum temperatures in July in its native range were 25.7 °C and 14.3 °C, were 24.3 °C and 13.3 °C in August, and were 19.1 °C and 8.9 °C in September (http://www.weatherdt.com/ (accessed on 23 May 2022)) (Appendix A). Therefore, a gradual decrease in temperature accompanies the maturing of the capitulum in its native range. Plants adapt to the climatic and environmental conditions of their original range in the long evolutionary process [20]. In Liaoning Province in China, the flowering time of *S. canadensis* is actually the same as that in its original Canadian home. This reflects the fact that the temperature from July to September in Liaoning province is similar to that in Canada (Appendix A). However, in Nanjing, the favorable temperature conditions for *S. canadensis* to flower occur from September to November (The mean maximum and minimum temperatures were 27.1 °C and 18.9 °C in September, 21.9 °C and 12.6 °C in October, and 15.5 °C and 6.2 °C in November) (Appendix A). In this study, the capitulum development of CN04, a hexaploid *S. canadensis* population in Nanjing, was in mid-September, and the development of spores occurred from mid-September to early October (Figure 5). Flowering and embryonic development began in mid-October, and mature seeds were formed in early November. The different times of sexual reproduction in the original and the invaded areas are probably the result of climatic adaptation [19,20]. Our previous research shows that introduced diploid populations of *S.canadensis* flowered from mid-June to early July, whereas introduced polyploids did not start flowering until October. Native polyploids also flowered later than the native diploids but earlier than introduced polyploids; the difference for flowering start between native diploids and polyploids was less extreme than those between introduced populations [2]. Nevertheless, introduced polyploids postponed flowering for at least a month compared to most native polyploids. This pre-adaptation led to polyploid *S. canadensis* producing significantly more viable seeds than the diploids, and polyploids from introduced populations produced more viable seeds than native polyploid counterparts. *S. canadensis* was a kind of short-day plant. The time of capitulum development in Canada and Nanjing differs by two months during the short-day stage. However, during the capitulum development in July, the day-length was 15 h/d, and the day-length in Nanjing in early September was 12 h/d. Therefore, the sexual reproduction process of *S. canadensis* is probably determined by temperature rather than absolute day-length.

The meiosis of microspore and megaspore is extremely sensitive to the external environment. Unfavorable conditions with respect to temperature and moisture are highly likely to cause meiosis failure [22,23,24], preventing the plants from carrying out normal sexual reproduction. The inherent temperature adaptability of *S. canadensis* is important in this context. The meiosis duration of various crops, such as sugar beet and wheat, is less than 30 h [25]. During our experiments, we found that the meiosis of the microspore and megaspore of the hexaploid *S. canadensis* population could last for about four to five days, respectively. This puts the meiosis at more risk of being affected by the environment. However, the microspore and megasporocyte in the capitulum observed in this study went through meiosis successfully. These results are consistent with previous studies indicating that embryo development and fruiting can adapt to the climate of the invasion site. It is likely that, after introduction to China, the delay of the capitulum development and the microspore/megasporogenesis of polyploid *S. canadensis* developed to avoid the high summer temperature. However, more experiments will be carried out in the future to support this supposition.

Research suggests that, during the flowering stage, rainfall could inhibit the activity of pollinators, the spread of pollen grains, and the elongation of pollen tubes [26], which in turn affects the normal fruiting of plants. In East China, there was less rainfall during the flowering period of *S. canadensis*, which benefited pollination and fertilization. It also made it easier for the seeds to leave the plants [27] because, without rain to cool the atmosphere, there was an upward current of warm air that took lightweight seeds to the upper air layers [28,29]. Then, in the winter months, the heavy monsoon (http://www.weatherdt.com/ (accessed on 23 May 2022)) provided excellent conditions for the long-distance propagation of the seeds. It seems that the climate in East China is conducive to the rapid spread and successful colonization of *S. canadensis*, and this result shows that the potential invasion range of the species could be predicted from local climate information. After researching the monthly average temperatures in different provinces in China, we find that, except for Qinghai, Tibet, and Hainan, most provinces have three consecutive months with similar average temperatures to the temperature from July to September in Ontario, Canada. Based on the analysis of climatic factors, *S. canadensis* could also have normal sexual reproduction in these areas. However, considering that *S. canadensis* has a physiological character that adapts well to wetness, the species is more likely to spread in North China, Northeast China and South China.

This study demonstrated that the anther wall of *S. canadensis* was dicotyledonous, and the mature pollen was a three-celled type, findings that are consistent with those of Pullaiah’s study [17]. We also found that *S. canadensis* has an anatropous ovule with a single integument, a thin nucellus, and a polygonum-type embryo sac. The integument tapetum starts from the megasporocyte stage, and the one-nucleate embryo sac stage is highly developed, features that are consistent with most of the reported embryological characteristics of the Compositae [30]. In our study, the female gametophytes were normally fertilized, and the plants could produce a large number of normal seeds. This finding was contrary to previous studies, which have suggested that female gametophytes do not undergo fertilization or double fertilization, and embryos cannot be produced normally, so the seeds are aborted [17]). However, other germination rate studies [31] and the embryological evidence both show that *S. canadensis* can be normally and sexually propagated. Population differences and the number of samples may have caused Pullaiah [17] to reach a different conclusion. Furthermore, the plant may behave differently in newly invaded areas, where it exhibits increased breeding capacity.

In this study, we investigated and established the relationship between the length of the capitulum and the sexual reproductive process in a hexaploid *S. canadensis* population, which was similar to many species in the Compositae [17,32]. It was observed that the tubular flower primordium was larger than the contemporary ligulate flower primordium within the capitulum. The sepals and petals of the tubular flower primordium also differentiated earlier than the ligulate flower primordium. Moreover, the panicle development and flowering of *S. canadensis* have obvious asynchrony, occurring one week apart. The flowering of different branches of the panicle was not synchronized, and the flowering in the bottom branch was about seven days later than that on the top. The fission of the pollen sac in the single capitulum of each tubular flower was also not synchronized. An asynchronized flowering strategy can not only enable the plant to complete the pollination and fertilization process successfully and ensure reproduction over an extended time-frame but can also reduce the negative effects of the environment on pollination and fertilization [19,33]. All of this helps to guarantee normal fruiting and promotes rapid invasion by spreading a large number of seeds.

Based on the results of this study, we have hypothesized a future treatment to inhibit the spread of *S. canadensis*, which is needed to develop a treatment that can successfully inhibit capitulum development of *S. canadensis*. In East China, where the invasion is most serious, this application can be used from mid-September to early October to inhibit capitulum development, prevent seed production, and ultimately control further spread. This study provides an appropriate basis for this treatment and opens up new horizons for the effective prevention and control of *S. canadensis* in the future.

## 4. Materials and Methods

### 4.1. Plant Materials and Sampling Method

A common garden field experiment was conducted at the Pailou Teaching and Research Station of Nanjing Agricultural University (32°02′ N, 118°37′ E), Nanjing, China [6]. The climate at this site is warm and humid, with an average annual precipitation of 1090 mm, and the mean annual temperature was 15 °C (lowest in February at 2.7 °C and highest in July at 28.1 °C). To prevent disturbance by animals and humans, the garden was established in a netted hoop house.

For the observation of the sexual reproduction process and phenology, an introduced *S. canadensis* population (population number: CN04; location: E: 118°53′25″ N: 32°01′25″; cytotypes: hexaploid) was chosen. In our previous research, all involved populations were identified as the same species, all belong to *S. canadensis* [2,34]. On 30 November 2005, seeds from 10 plants (ramets about 10 m from each other) in the population were collected (Appendix A). The seeds were mixed and were air-dried and stored in a refrigerator at 4 °C. Seeds were germinated in seedling trays filled with a 1:2 mixture of soil and sterilized compost in a greenhouse on 5 December, 2006. After germination (6 March, 2007), a total of 210 small plants (3–5 leaves) were transplanted in 5 L plastic pots (3 plants per pot) with the same growing medium and were grown in a netted hoop house.

The experiment was conducted in 2007 and 2008. For each year, from 01 September to 10 October, plants at the same growth stage were selected every day during the sexual reproduction process, and 1 top capitulum was sampled from each of 20 randomly selected individual plants. The lengths of the capitula were measured under a stereo microscope (Olympus SZX7, Tokyo, Japan) with electronic digital calipers (B229-090, Shanghai Measuring and Cutting Tool Factory, Shanghai, China).

### 4.2. Observation of Capitulum Development

In order to record morphological variations of the capitulum during capitulum development, 10 capitula were peeled off with a dissecting needle and a scalpel to remove the periclinium, double-fixed with glutaraldehyde and osmic acid, dehydrated by ethanol, dried, sprayed with gold, and then observed and photographed by scanning electron microscopy (Hitachi S-3500N). The other 10 capitula were fixed with FAA fixative, prepared by conventional paraffin sectioning, sliced into a thickness of 6 μm, stained with hematoxylin-Fast Green FCF [35], and sealed with vegetable gum. Observation and photography were conducted with an optical microscope (Olympus BH-2, Tokyo, Japan).

### 4.3. Observation of Megasporogenesis and Microsporogenesis, the Development of Male and Female Gametophytes, and Embryonic Development

In order to record the morphological variation during megasporogenesis, microsporogenesis, and embryonic development, the samples were put in FAA fixative and made into a conventional paraffin section (AO 820 rotary microtome, AO scientific instrument, Shanghai, China), which was sliced into a thickness of 10 μm and stained with hematoxylin-Fast Green FCF [35]. The samples were then sealed with gum. Observation and photography were conducted with an optical microscope (Olympus BH-2). 

### 4.4. Acquisition of Meteorological Data

The meteorological information (temperature, wind force, and rainfall) were obtained through the databases of WorldClim (https://www.worldclim.org/ (accessed on 23 May 2022)).

## Figures and Tables

**Figure 1 plants-11-02073-f001:**
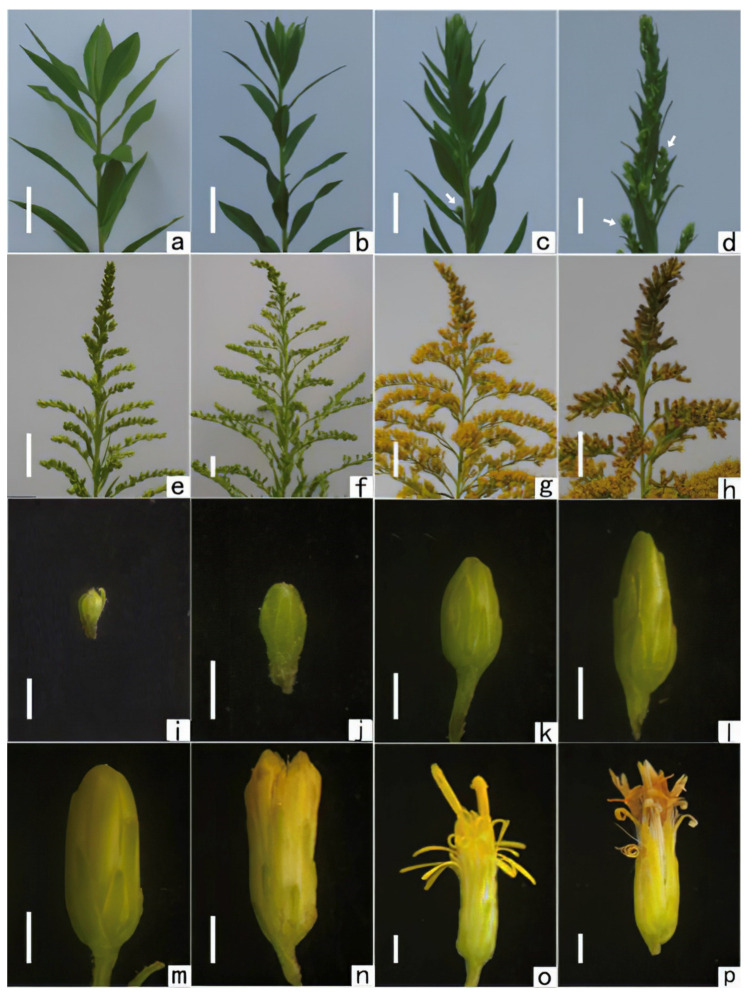
Plant and capitulum morphologies during the reproductive growth of *S. canadensis* (population number: CN04). (**a**) Plant in the vegetative growth stage. (**b**) Plant in the late vegetative growth stage. (**c**) Capitulum bud (arrowhead) occurrence in the axillary buds on the short branches. (**d**) End of capitulum development (arrowheads) on the top of the plant. (**e**) Developing panicle. (**f**) Completely developed panicle. (**g**) Flowering panicle. (**h**) Withering panicle. (**i**) Capitulum during capitulum development. (**j**) Capitulum with a completely differentiated capitulum bud. (**k**) Capitulum during the microsporocyte stage. (**l**) Capitulum during the microspores meiosis stage. (**m**) Capitulum during the developing-male-gametophyte stage. (**n**) Capitulum with mature pollen. (**o**) Flowering capitulum. (**p**) Withering capitulum. Scale bars: a–d = 2 cm, e–h = 5 cm, i–p = 1 cm.

**Figure 2 plants-11-02073-f002:**
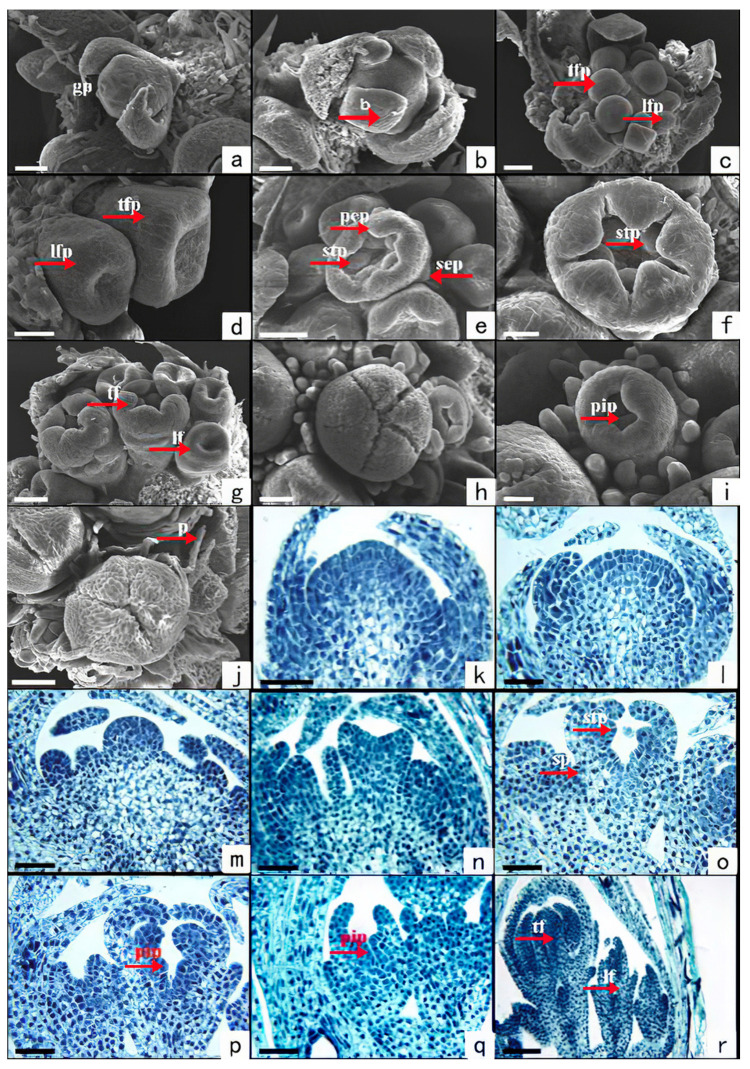
Capitulum development of *S. canadensis* (population number: CN04). Observed and photographed by scanning electron microscopy (**a**–**j**) and optical microscope (**k**–**r**). (**a**,**k**) The pre-differentiation stage. (**b**,**l**) The periclinium differentiation stage. (**c**,**m**) The floret primordia differentiation stage. (**d**,**n**) The corolla primordium differentiation stage. (**e**,**o**) Differentiation of the calyx and the stamen primordium. (**f**) Stamen primordia enlarge. (**g**) The second sepals of the tubular flower and the sepals of the ligulate flower form. (**h**,**p**) Closed corolla of a tubular flower and stamen differentiation. (**i**,**q**) Stamen differentiation of a ligulate flower. (**j**,**r**) End of the differentiation. Scale bars: a–c = 50 μm, d =25 μm, e = 50 μm, f = 25 μm, g–h = 50 μm, i = 25 μm, j = 100 μm, k–q = 50 μm, r = 100 μm. gp, Growing point; b, bract; tfp, tubular flower primordium; lfp, ligulate flower primordium; pep, petal primordium; stp, stamen primordium; sep, sepal primordium; pip, pistil primordium; p, pappus; tf, tubular flower; lf, ligulate flower.

**Figure 3 plants-11-02073-f003:**
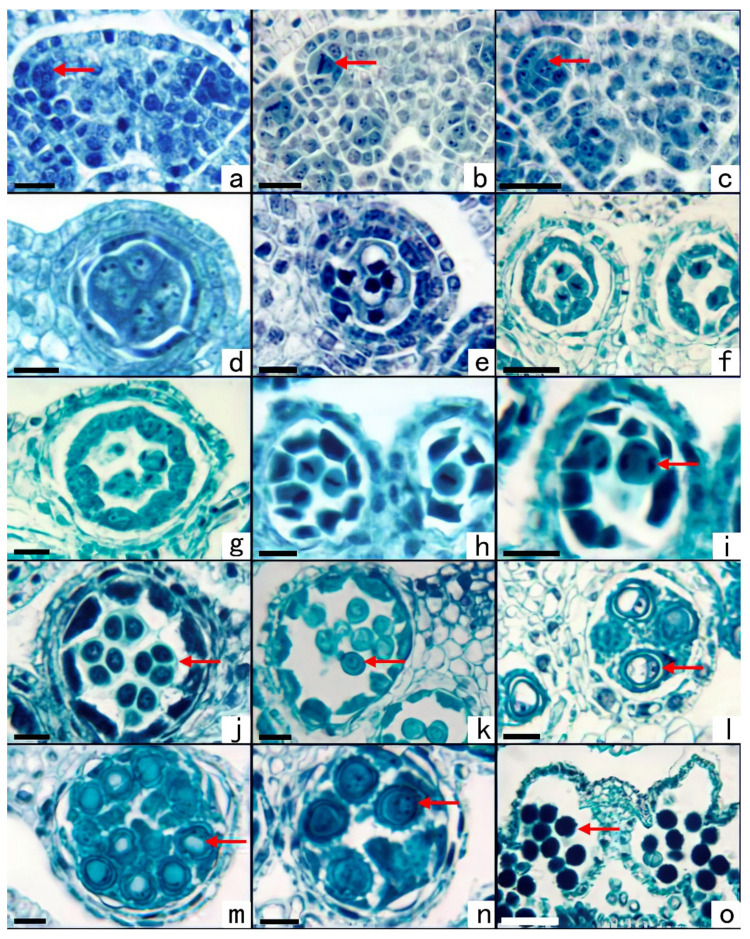
Microsporogenesis and male gametophyte development of *S. canadensis* (population number: CN04). Section stained with hematoxylin-Fast Green FCF. (**a**). Archesporial cell. (**b**) Archesporial cell in the metaphase of periclinal division. (**c**) Primary parietal cells and primary sporogenous cells. (**d**) Microsporocytes and four-layer walls. (**e**) Prophase Ⅰ. (**f**) Metaphase Ⅰ. (**g**) Anaphase Ⅰ. (**h**) Metaphase Ⅱ, wherein the tapetum begin to dissolve. (**i**) Anaphase Ⅱ. (**j**) Tetrahedral tetrad. (**k**) Early one-nucleate microspore, nucleus in the middle of the cell. (**l**) One-nucleate side-stage microspore, and the periplasmodium flows into the anther chamber. (**m**) 2-celled pollen. (**n**) 3-celled pollen. (**o**) Mature pollen escapes from the pollen sac. Scale bars: a–c = 25 μm, d–e = 12.5μm, f = 25 μm, g–n = 12.5 μm, o = 50 μm.

**Figure 4 plants-11-02073-f004:**
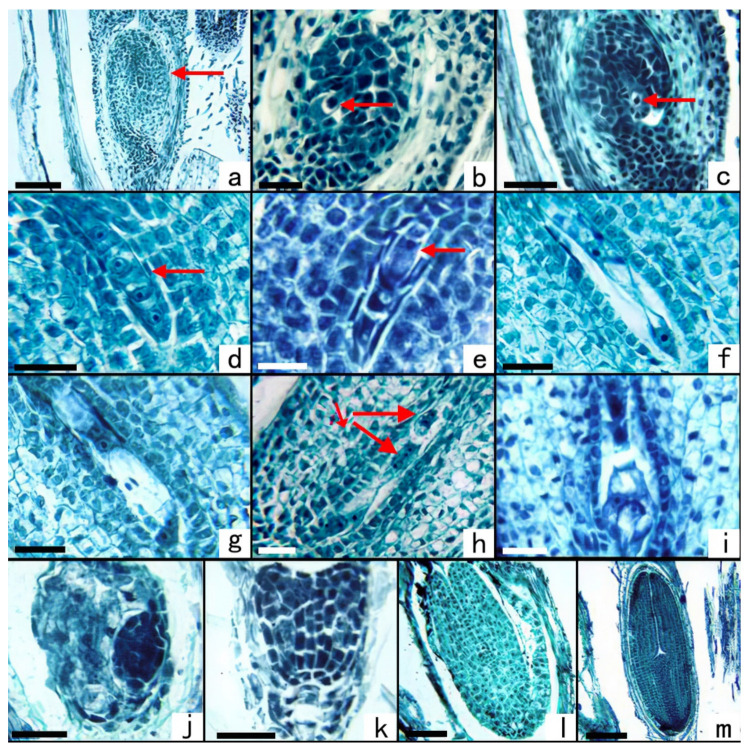
Megasporogenesis, female gametophyte, and embryonic development of *S. canadensis* (population number: CN04). Section stained with hematoxylin-Fast Green FCF. (**a**) Anatropous ovule. (**b**) Archesporial cell. (**c**) Megasporocyte. (**d**) Linear megaspore tetrad. (**e**) The megaspore at the chalazal end became a functional megaspore and developed into a one-nucleate embryo sac. (**f**) Two-nucleate embryo sac. (**g**) Four-nucleate embryo sac. (**h**) Eight-nucleate embryo sac, showing one synergid, one egg cell, one polar nuclear, and two multinucleated antipodal cells arrayed in tandem. (**i**) Mature embryo sac. (**j**) Globular embryo. (**k**) Heart-shaped embryo. (**l**) Torpedo-shaped embryo. (**m**) Mature embryo A: antipodal cells. Scale bars: a = 100 μm, b = 25 μm, c = 50 μm, d–h = 25 μm, i–j = 50 μm, k–m = 100 μm.

**Figure 5 plants-11-02073-f005:**
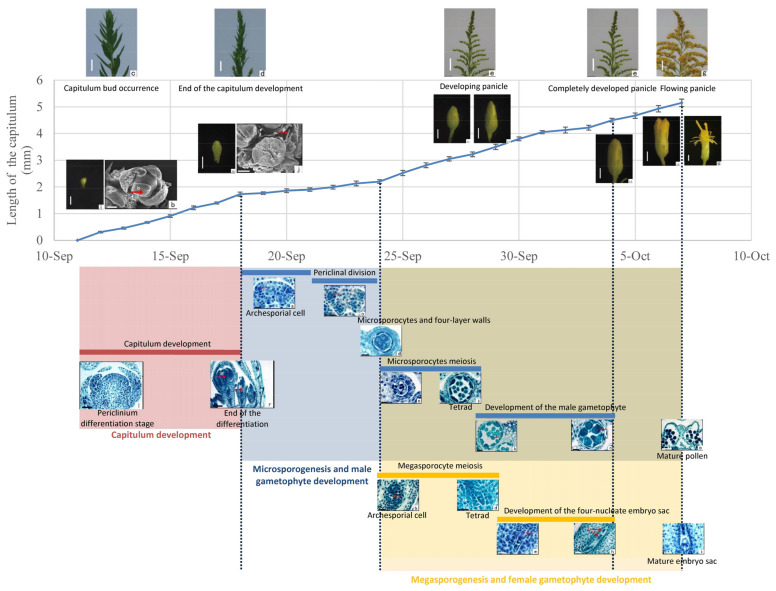
Reproductive calendar of *S. canadensis* (population number: CN04) representing the main developmental stages. Colored lines indicate the duration of the corresponding stage.

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
