# Peer review of "Capitulum Development and Gametophyte Ontogeny: Histological Insight into the Reproductive Process of a Hexaploidy Population of Solidago canadensis in China"

_plants, 2022, doi:10.3390/plants11152073_

Round 1
Reviewer 1 Report
The idea for this research is very interesting and important as can explain more about the invasive potential of Canadian goldenrod.
My main comments:
1. Referring to the aim formulated in the manuscript, the results content is not clear. For example, in the aim, it is written that a particular emphasis will be put on the relationship between development in the context of climatic factors in original and invaded areas. I don't see any results that would clearly refer to this ambitious aim.
Also, I understand difficulty of studying S. canadensis that has inflorescens that do not mature in the same time. That is why I would appreciate mentioning in Materials and Methods which part of the flowers were the capitula taken from - totally random or not?
In fact, there is no reference regarding the population source for the results in Figures 1-4. Is it Chinese or Canadian?
2. In the supplementary material there are several different populations of S. canadensis mentioned, that were studied, but I can't see these results in the manuscript.
3. Also, how was the species S. canadensis confirmed botanically. There is no mention of this crucial taxonomic part of this study.
4. Figure 5 - is very important, but does it relate to the Chinese population or Canadian? It is not explained.
5. Materials and methods - is population CN04 from the native range? Is it of native origin? It is not explained.
Author Response
Response to Reviewer 1 Comments
- Referring to the aim formulated in the manuscript, the results content is not clear. For example, in the aim, it is written that a particular emphasis will be put on the relationship between development in the context of climatic factors in original and invaded areas. I don't see any results that would clearly refer to this ambitious aim.
Response: Thanks for the reviewer’s comments and kind suggestions. In previous studies (Cheng et al., 2020), we have revealed that introduced populations of S.canadensis flowered from mid-June to early July, whereas introduced polyploids did not start flowering until October. We also revealed the differentiation of flowering time in the context of climatic factors in different regions. This pre-adaptation led to introduced polyploid S. canadensis produced more viable seeds than native polyploid counterparts. The reproductive process is more sensitive to the environment than the vegetative growth, it was therefore extremely important to successful invasion of S.canadensis. However, there is no comprehensive study on complete process of sexual reproduction in S. canadensis for reference. Therefore, the purpose of this study is to use the introduced population as the research object to reveal capitulum development and embryo development after the invasion and colonization of the species under the climatic conditions of the invasion site, so as to provide a theoretical basis for effectively inhibiting its reproductive process and preventing proliferation in the future. According to reviewer advices, the sentence has been improved as” The purpose is to reveal the complete process of sexual reproduction of the species after invasion and colonization under the climatic conditions of the invasion site, and to provide a theoretical basis for preventing and controlling the spread of the species.” And the detailed revision can be found in Line 75-77, Page 2.
Also, I understand difficulty of studying S. canadensis that has inflorescens that do not mature in the same time. That is why I would appreciate mentioning in Materials and Methods which part of the flowers were the capitula taken from - totally random or not?
Response: Thanks for the reviewer’s kind suggestion. During sampling, for 20 randomly selected samples, we selected a capitulum at the top of the inflorescence axis. This point can be found in Line 433-434, Page 15. And L438-445,Page 15-16.
In fact, there is no reference regarding the population source for the results in Figures 1-4. Is it Chinese or Canadian?
Response: Thanks for question. All experiences in this study is based on the invasive population CN04. According to his/her advices, the population information has been added in this revision version. Please refer to the article for details.
- In the supplementary material there are several different populations of S. canadensis
Response: This study cites the conclusion of our previous study (Cheng ei al., 2021) that the flowering time of S. canadensis will differentiate with the climate conditions in different regions. In the supplementary material, in order to specify the temperature changes in different regions, we selected the regions where we had collected population samples or conducted common garden experiments to obtain the temperature data. Therefore, there are two North American populations (CA11, CA20), one Chinese populations (CN04) and one regions (Liaoning). The sexual reproduction and seed germination of these populations have been explained in previous studies, in this one we only focuses on CN04 population.
- Also, how was the species S. canadensis confirmed botanically. There is no mention of this crucial taxonomic part of this study.
Response: It's a very valuable and professional question. In our previous research, two molecular markers ITS (Table S1) and psbA-trnH (Table S2) showed all plants from 152 selected S. canadensis populations have identical sequences, indicating that all involved populations are the same species and, hence, all involeved polyploidy populations are S. canadensis.
Table 1: Test of ITS sequences of Solidago spp. blasted in NCBI database
|
Population codes |
Accession number from Genebank |
Species1 |
Query cover |
Homology comparison to S. canadensis |
Variable sites |
|
|
CN04 |
HQ142590.1 |
S. canadensis |
100% |
100% |
||
|
S. canadensis |
HQ142590.1 |
S. canadensis |
100% |
100% |
||
|
S. gigantea |
HQ142593.1 |
100% |
99.75% |
583 (C→T), 627 (T→C) |
||
|
S. decurrens |
JN204176.1 |
100% |
99.21 |
627 (T→C), 668 (G→A), 724-726 (GTG→TCA) |
||
|
S. simplex |
HQ142576.1 |
100% |
99.38% |
627 (T→C), 724-726 (GTG→TCA) |
||
|
S. flexicaulis |
HQ142582.1 |
98% |
99.26% |
132 (G→A), 257 (G→A), 627 (T→C), 724-726 (GTG→TCA) |
||
|
S. juncea |
HQ142584.1 |
99% |
99.63% |
132 (G→A), 668 (G→A) |
||
|
S. ptarmicoides |
EU125364.1 |
100% |
99.04% |
132 (G→A), 319 (C→A), 627 (T→C), 724-726 (GTG→TCA) |
||
|
S. petiolaris |
HQ142577.1 |
98% |
99.38% |
175 (C→T), 627 (T→C), 724-726 (GTG→TCA) |
||
|
S. nitida |
HQ142598.1 |
98% |
99.26% |
627 (T→C), 724-726 (GTG→TCA) |
||
|
S. ohioensis |
HQ142599.1 |
100% |
99.38% |
583 (C→T), 627 (T→C), 724-726 (GTG→TCA) |
||
|
S. virgaurea |
KU872548.1 |
|
99% |
98.67% |
503 (C→T), 550 (C→A), 627 (T→C), 668 (G→A), 724-726 (GTG→TCA), 797 (-→C), 801 (-→A) |
Note: S. gigantean, S. decurrens, S. simplex, S. flexicaulis, S. ptarmicoides, S. petiolaris, S. nitida, S. ohioensis and S. virgaurea in the table were downloaded from the GenBank and compared with Solidago canadensis L. (accession numbers HQ142591.1). "-" indicates base deletion.
Table 2: Test of psbA-trnH intergenic spacer sequences of Solidago spp. blasted in NCBI database
|
Population codes |
Accession number from Genebank |
Species |
Query cover |
Homology comparison to Solidago canadensis |
Variable sites |
|
|
CN04 |
KX214929.1 |
S. canadensis |
91% |
100% |
||
|
S. canadensis |
KX214929.1 |
|||||
|
S. gigantea |
EU337333.1 |
91% |
96.79% |
74-79 (-------→ATAGTA) |
||
|
S. decurrens |
KX346952.2 |
85% |
99.65% |
119 (-→T) |
||
|
S. simplex |
HQ142543.1 |
98% |
97.38% |
68-73 (------→TATTAT) |
||
|
S. flexicaulis |
EU750565.1 |
83% |
100% |
|||
|
S. juncea |
HQ142560.1 |
98% |
98.08% |
25-28 (TAGT→----) |
||
|
S. ptarmicoides |
HQ142569.1 |
98% |
98.08% |
33-37 (TTATA→-----) |
||
|
S. petiolaris |
HQ142544.1 |
98% |
97.39% |
43-49 (-------→TATATAT) |
||
|
S. nitida |
HQ142567.1 |
98% |
98.08% |
86-90 (ATACT→-----) |
||
|
S. ohioensis |
HQ142568.1 |
98% |
98.08% |
33-37 (TTATA→-----) |
||
|
S. virgaurea |
EU337704.1 |
|
99% |
98.51% |
33-37 (TTATA→-----) |
Note: S. gigantean, S. decurrens, S. simplex, S. flexicaulis, S. ptarmicoides, S. petiolaris, S. nitida, S. ohioensis and S. virgaurea in the table were downloaded from the GenBank and compared with Solidago canadensis L. (accession numbers KX214929.1). "-" indicates base deletion.
- Figure 5 - is very important, but does it relate to the Chinese population or Canadian? It is not explained.
Response: Thanks for the reviewer’s kind suggestion. The population CN04 is from introduced range, this information was given in this revised version and the detailed revision can be found in Line 269, Page 12.
- Materials and methods - is population CN04 from the native range? Is it of native origin? It is not explained.
Response: The population CN04 is from introduced range, this information was given in this revised version and the detailed revision can be found in the article.
Reference
- Cheng, J.L.; Yang, X.H.; Xue, L.F.; Yao, B.B.; Lu, H.; Tian, Z.S.; Li, J. Zhou, X.; Zhang, Y.; Zia, U.H.M., Wu. S., Song, X.L.; Hu, S.J., and Qiang, S. Polyploidization contributes to evolution of competitive ability: a long term common garden study on the invasive Solidago canadensis in China. 2020, 129.
- Cheng, J, Li, J, Zhang, Z, Lu, H, Chen, G, Yao, B, Dong, Y, Ma, L, Yuan, X, Xu, J, Zhang, Y, Dai, W, Yang, X., Xue, L, Zhang, Y, Zhang, C, Mauricio, R., Peng, G, Hu, S, Valverde, B. E, Song, X, Li, Y, Stift, M, Qiang, S. (2021). Autopolyploidy-driven range expansion of a temperate-originated plant to pan-tropic under global change. Ecological Monographs 91(2):e01445
- Semple, J. C. 2016. An intuitive phylogeny and summary of chromosome number variation in the goldenrod genus Solidago (Asteraceae: Astereae). Phytoneuron, 32, 1-9.
Reviewer 2 Report
Manuscript studies in detail the stages of the Solidago canadensis reproductive process to prepare a reproductive calendar showing the key reproductive events. The authors claim that their results open up new horizons for effective prevention and control of the spread of this weed. But I think this particular statement is a bit of a long shot: The manuscript describes the stages of the reproductive process in great detail, but it did not deduce or discuss extensively, why potential management should aim to suppress capitulum development or how this study contributes to it, except finding out when this happens.
L403–L407: Particularly this paragraph seems a bit long shot to me. If you know that there is a treatment that successfully inhibits capitulum growth and that it is effective (at least for other species), these citations should be used to support your ideas presented in this paragraph. If there is no such treatment, then it is already a very long shot when you claim that such a hypothetical treatment should be used at a time you specify. Thus, the study, although undoubtedly beneficial, does not open "new horizons for effective prevention of S. canadensis control".
Minor corrections:
L21: correct „successful invaded“ to „successful invasion“.
L29: Because I'm from Europe, "Midwest of Europe" sounds very ridiculous to me - I don't know what the authors meant, but "all over Europe" would be good enough, because a quick look at the gbif database shows that this species is widespread throughout the "The old continent".
L44: „is grown“ means, that it is intentionally reared with other species – is this what the authors wanted to say, or did that mean that it simply grows with other species?
L45: add a space between “colonization” and “(Cheng et al. 2020)”.
Figure 3. The exact same image is used as in Figure 2. Did the authors want to use the same image or it is a mistake - if they wanted to, the picture lacks a description of 3p, 3q, and 3r in Figure 3, or oppositely some subpictures in Figure 3 are redundant having no description.
Figure 5. S. canadensis is not italic in the label.
L277: „wind greater than Force 5“ – please specify in kilometers per hour or in some other SI system unit.
L309-L310: The names of the months should begin with a capital letter.
Author Response
Response to Reviewer 2 Comments
Manuscript studies in detail the stages of the Solidago canadensis reproductive process to prepare a reproductive calendar showing the key reproductive events. The authors claim that their results open up new horizons for effective prevention and control of the spread of this weed. But I think this particular statement is a bit of a long shot: The manuscript describes the stages of the reproductive process in great detail, but it did not deduce or discuss extensively, why potential management should aim to suppress capitulum development or how this study contributes to it, except finding out when this happens.
L403–L407: Particularly this paragraph seems a bit long shot to me. If you know that there is a treatment that successfully inhibits capitulum growth and that it is effective (at least for other species), these citations should be used to support your ideas presented in this paragraph. If there is no such treatment, then it is already a very long shot when you claim that such a hypothetical treatment should be used at a time you specify. Thus, the study, although undoubtedly beneficial, does not open "new horizons for effective prevention of S. canadensis control".
Response: Thanks for the valuable comments. In fact, our laboratory has developed a chemical preparation that can effectively inhibit inflorescence development of S.canadensis, which has not been published yet, because it involves patent issues. Our research showed this “Flower bud inhibitors” can effectively inhibit the development of inflorescence, reduce the amount of seed setting, and then inhibit the spread of seeds. This prevention and control method is of great significance to control the sexual reproduction of S. canadensis. These contents may be published later.
Minor corrections:
L21: correct „successful invaded“ to „successful invasion“.
Response: Correction has been made.
L29: Because I'm from Europe, "Midwest of Europe" sounds very ridiculous to me - I don't know what the authors meant, but "all over Europe" would be good enough, because a quick look at the gbif database shows that this species is widespread throughout the "The old continent".
Response: Correction has been made.
L44: „is grown“ means, that it is intentionally reared with other species – is this what the authors wanted to say, or did that mean that it simply grows with other species?
Response: Thanks for the questions. In our previous study, we performed a common garden field experiment by planting different geo-cytotype populations of S. canadensis with local species, and led them compete under none artificial interference conditions, in order to verified whether there are differences in biodiversity, species composition and community succession among different experiment plots. The detailed information can be found in the original paper (Cheng et al. 2020).
L45: add a space between “colonization” and “(Cheng et al. 2020)”.
Response: Correction has been made.
Figure 3. The exact same image is used as in Figure 2. Did the authors want to use the same image or it is a mistake - if they wanted to, the picture lacks a description of 3p, 3q, and 3r in Figure 3, or oppositely some subpictures in Figure 3 are redundant having no description.
Response: We are sorry for the mistake. The correct figure 3 has been replaced, and the detailed revision can be found in Line 175, Page 9.
Figure 5. S. canadensis is not italic in the label.
Response: Correction has been made.
L277: „wind greater than Force 5“ – please specify in kilometers per hour or in some other SI system unit.
Response: Thanks for the reviewer’s kind suggestion. Correction has been made. and the detailed revision can be found in Line 283, Page 12.
Reference
- Cheng, J.L.; Yang, X.H.; Xue, L.F.; Yao, B.B.; Lu, H.; Tian, Z.S.; Li, J. Zhou, X.; Zhang, Y.; Zia, U.H.M., Wu. S., Song, X.L.; Hu, S.J., and Qiang, S. Polyploidization contributes to evolution of competitive ability: a long term common garden study on the invasive Solidago canadensis in China. 2020, 129.
- Cheng, J, Li, J, Zhang, Z, Lu, H, Chen, G, Yao, B, Dong, Y, Ma, L, Yuan, X, Xu, J, Zhang, Y, Dai, W, Yang, X., Xue, L, Zhang, Y, Zhang, C, Mauricio, R., Peng, G, Hu, S, Valverde, B. E, Song, X, Li, Y, Stift, M, Qiang, S. (2021). Autopolyploidy-driven range expansion of a temperate-originated plant to pan-tropic under global change. Ecological Monographs 91(2):e01445
- Semple, J. C. 2016. An intuitive phylogeny and summary of chromosome number variation in the goldenrod genus Solidago (Asteraceae: Astereae). Phytoneuron, 32, 1-9.
Round 2
Reviewer 1 Report
Still, there are several references to the Canadian population of S. canadensis in the Results and Materials and Methods, but there are no results for the native Canadian range population presented in the manuscript. That must be clarified before publishing.
Also, you explained the botanical taxonomy in the responses to the Reviewer, but no reference to the source has been presented in the corresponding chapter of your manuscript. That also must be clarified.
Author Response
Still, there are several references to the Canadian population of S. canadensis in the Results and Materials and Methods, but there are no results for the native Canadian range population presented in the manuscript. That must be clarified before publishing.
Response: Thanks for the reviewer’s comments and kind suggestions. This part has been corrected, and the detailed revision can be found in Line 265-268, Page 10; Line 313-315, Page 12 and Line 460, Page 15.
Also, you explained the botanical taxonomy in the responses to the Reviewer, but no reference to the source has been presented in the corresponding chapter of your manuscript. That also must be clarified.
Response: Thanks for the reviewer’s kind suggestion. According to reviewer advices, this part has been improved, and the detailed revision can be found in Line 426-427, Page 14.
Round 3
Reviewer 1 Report
I find the paper ready to be published now.
Author Response
Major points
Some important quantitative data should be shown in Figures or Tables.
Response: Thanks for the reviewer’s kind suggestion. The quantitative data you mentioned, such as the length of the capitulum and the duration of each developmental stages was shown in figure 5.
Using tracking system might make Figures and/or Tables smaller, which could make the review very difficult. The manuscript file without tracking system is also necessary to finalize this manuscript.
Response: Correction has been made.
Mainor points
Figure 1: c an arrowhead to indicate the buds is necessary. d: arrowheads to indicate capitulum are necessary. The author could add arrowheads to other figures for the readers unfamiliar with plant morphology.
Response: Correction has been made.
p4L161 "flowing" should be replaced by "flowering".
Response: Correction has been made.
Figure 2. Explanation of microscopic technique should be added briefly in the caption section.
Why the term Population number CN04 is typed in red?
Response: Correction has been made.
Figures 3 and 4. Staining technique should be added to the caption section.
Response: Correction has been made.
P8L262 "syner.gid" should be replaced by "synergid".
Response: Correction has been made.
Figure 5 should be magnified, or review process is impossible.
Response: Due to the problem of word version, the pasted pictures will be compressed. Figure 5 and other figures have been uploaded separately as attachment.
P11L353"sunshine duration" could be replaced by "day-length".
Response: Correction has been made.
P12: Information of chromosome number (ploidy) of the plant material in this study should be added. DNA polymorphism information shown in the response to review seems interesting, and could be cited in the discussion section and/or materials and methods section.
Response: Thanks for the reviewer’s kind suggestion. Correction has been made. This point can be found in Line 417-418, Page 12
About the chromosome number (ploidy) that you mentioned, we have discussed this topic in several previous studies, (Lu et al., 2020; Cheng et al., 2020,2021; Yang et al., 2021) clarified that the impact of polyploidization of S. canadensis on its invasion ability. In this paper we didn't involve this part in order to prevent data duplicate, thus the relevant content is not described in detail.
Lu, H., Xue, L., Cheng, J., Yang, XH, Xie, H., Song, X., & Qiang, S. (2020). Polyploidization‐driven differentiation of freezing tolerance in Solidago canadensis. Plant, Cell & Environment. DOI: 10.1111/pce.13745.
Cheng, J*, Yang, XH*, Xue, L., Yao, B., Lu, H., Tian, Z., Li, J., Zhou, X., Zhang, Y., Zia Ul Haq, Wu, S., Song, X., Hu, S., Qiang, S. (2020). Polyploidization contributes to evolution of competitive ability: a long term common garden study on the invasive Solidago canadensis in China. Oikos 129: 700-713.
Cheng, J.L.; Li, J.; Zhang, Z,; Lu, H,; Chen, G,Q,; Yao,B,B,; Dong, Y,X,; Ma, L,; and Qiang, S. Autopolyploidy-driven range expansion of a temperate-originated plant to pan-tropic under global change. Ecological Monographs, 2021, 91(2):e01445.
Yang, XH, Cheng, J., Yao, B., Lu, H., Zhang, Y., Xu, J., Song, X., Sowndhararajan, K., Qiang, S. (2021). Polyploidy-promoted phenolic metabolism confers the increased competitive ability of Solidago canadensis. Oikos. 130: 1014–1025.
I afraid that the reference 6 might not be present.
Response: This reference has been removed
I am afraid that an academic journal named Journal of Biomathematics might not be present.
Response: This reference has been removed
Page information of reference 16 is lacking.
Response: Correction has been made.
The reference 24 was not published in J. Plant Res. in 1976.
Response: Correction has been made.
I afraid that the reference 34 might not be present.
Response: This reference has been removed
